# Safety and efficacy of N-acetylcysteine in hospitalized patients with HIV-associated tuberculosis: An open-label, randomized, phase II trial (RIPENACTB Study)

Izabella Picinin Safe[1,2], Marcus Vinícius Guimarães Lacerda[1,2,3]*, Vitoria Silva Printes[1], Adriana Ferreira Praia Marins[1], Amanda Lia Rebelo Rabelo[2], Amanda Araújo Costa[1], Michel Araújo Tavares[4], Jaquelane Silva Jesus[1], Alexandra Brito Souza[2], Francisco Beraldi-Magalhães[2], Cynthia Pessoa Neves[1,2], Wuelton Marcelo Monteiro[1,2], Vanderson Souza Sampaio[1,2], Eduardo P. Amaral[5], Renata Spener Gomes[2], Bruno B. Andrade[6,7,8,9,10,11,12]☯, Marcelo Cordeiro-Santos[1,2]☯

1 Fundação de Medicina Tropical Dr. Heitor Vieira Dourado, Manaus, Brazil, 2 Programa de Pós-Graduação em Medicina Tropical, Universidade do Estado do Amazonas, Manaus, Brazil, 3 Instituto Leônidas & Maria Deane, Fiocruz, Manaus, Brazil, 4 Universidade Federal do Amazonas, Manaus, Brazil, 5 Immunobiology Section, National Institutes of Allergy and Infectious Diseases, National Institutes of Health, Bethesda, Maryland, United States of America, 6 Laboratório de Inflamação e Biomarcadores, Instituto Gonçalo Moniz, Fundação Oswaldo Cruz (FIOCRUZ), Salvador, Brazil, 7 Multinational Organization Network Sponsoring Translational and Epidemiological Research, (MONSTER) Initiative, Salvador, Brazil, 8 Universidade Salvador (UNIFACS), Laureate Universities, Salvador, Brazil, 9 Curso de Medicina, Faculdade de Tecnologia e Ciências, Salvador, Brazil, 10 Escola Bahiana de Medicina e Saúde Pública (EBMSP), Salvador, Brazil, 11 Wellcome Centre for Infectious Diseases Research in Africa (CIDRI-Africa), Institute of Infectious Disease and Molecular Medicine, University of Cape Town (UCT), Cape Town, South Africa, 12 Division of Infectious Diseases, Department of Medicine, Vanderbilt University School of Medicine, Nashville, TN, United States of America

☯ These authors contributed equally to this work.
* marcuslacerda.br@gmail.com

## Abstract

Despite the availability of effective antimicrobials, tuberculosis (TB) is still a serious health threat. Mortality is even higher in people living with HIV who are diagnosed with TB. New therapies are needed to shorten the time required to cure TB and decrease fatality rates in this population. N-acetylcysteine (NAC) is a glutathione precursor and has shown recently in experimental setting to present *in vitro* and *in vivo* anti-mycobacterial activity. We test the hypothesis that NAC is safe, well tolerated and secondarily efficacious as adjunctive anti-TB therapy in hospitalized individuals with HIV-associated TB. Patients were enrolled sequentially in a tertiary care center, in the Brazilian Amazon. We performed a randomized, parallel group, single-center, open study trial of two arms, in hospitalized patients over 18 years of age, with microbiologically confirmed pulmonary TB in HIV: one with rifampicin, isoniazid, pyrazinamide and ethambutol at standard doses (Control Group), and a second in which NAC 600 mg bid for eight weeks was added (NAC Group). A total of 21 and 18 patients were enrolled to the Control Group and NAC Group, respectively. Adverse event rates were similar in the two arms. Our findings suggest that in the more critical population of hospitalized patients with HIV-associated TB, the use of NAC was not unsafe, despite the low

**Data Availability Statement:** All relevant data are within the manuscript and its Supporting Information files.

**Funding:** This work was supported by Cooperação Interfederativa do Amazonas (Interfam), funded by the Brazilian Ministry of Health. MVGL, WMM and BBA are CNPq fellows.

**Competing interests:** The authors have declared that no competing interests exist.

sample size, and a potential impact on faster negative cultures needs to be further explored in larger studies.

## Introduction

Worldwide, tuberculosis (TB) is one of the top 10 causes of death. Despite significant efforts to control the disease, World Health Organization (WHO) estimates that 250,000 people living with HIV died in 2018 due to TB [1]. Treatment scheme of TB in people with HIV is the same as in HIV-negative patients. The recommended regimen for drug-susceptible disease is a combination of Rifampicin, Isoniazid, Pyrazinamide, Ethambutol (RIPE) for 2 months, followed by at least 4 months of rifampicin and isoniazid [2]. Although it is a curable and treatable disease, TB is the leading cause of death (40%), admission to hospital (18%), and in-hospital death (25%) in people living with HIV (PLWH) [3,4].

Glutathione (GSH) is the main nonprotein thiol responsible for cellular homeostasis and maintenance of the cellular redox balance [5]. HIV infection is associated with increased oxidative stress (OS). Intracellular GSH levels in macrophages of HIV individuals are compromised, contributing to the loss of innate immune function observed in these patients and an increase in the growth of *Mycobacterium tuberculosis* (Mtb) [6]. Agents that assist in the restoration of GSH levels in macrophages isolated from individuals with HIV infection promote better control of Mtb [7]. N-acetylcysteine (NAC), a GSH precursor, is an agent that restores GSH levels. T lymphocytes derived from HIV infected individuals are deficient in GSH, and this deficiency correlates with decreased levels of Th1 cytokines and enhanced growth of Mtb inside human macrophages [8]. NAC was shown to tailor macrophages to induce enhanced Th1 response that may be helpful to control TB [9].

NAC is included in the list of essential medicines of WHO [10]. It is widely used in patients with a wide range of respiratory diseases due to its mucolytic and anti-oxidant activities, making it attractive as a potential chronic obstructive pulmonary disease therapy [11]. NAC potentially protects against anti-TB drug-induced hepatotoxicity in individuals with TB without HIV [12]. NAC treatment in Mtb-infected human macrophages resulted in a decrease of oxidative stress and enhanced anti-mycobacterial activity [13]. In a model of Mtb infection of mice, NAC treatment resulted in diminished mycobacterial loads in lungs [13], highlighting the therapeutic potential of this drug.

NAC as an adjuvant appears to be an effective agent in terms of early bacteriological and radiological improvement in treatment of pulmonary TB [14]. However, to our knowledge, no evidence exists for patients with HIV-associated TB, and especially in those more complicated cases requiring hospitalization. Such clinical trial play innovative and strategic role in WHO New Global Elimination Tuberculosis Strategy (Pillar III research strategy) [15].

## Methods

### Ethics

The study was approved by the Ethics Review Committee of *Fundação de Medicina Tropical Dr Heitor Vieira Dourado* (CAAE 60219916.5.0000.0005). Written informed consent was obtained from all participants (or relatives in case of unconscious patients), after detailed information about the study protocol was given.

## Study design

RIPENACTB Study was an open-label, single center, randomized, phase II trial to test whether NAC-containing treatment regimen was as safe as the standard regimen for TB treatment in hospitalized patients with HIV, besides exploring efficacy upon respiratory sample culture conversion. The study was conducted at *Fundação de Medicina Tropical Dr Heitor Vieira Dourado* (FMT-HVD), a tertiary care reference institution for coinfection TB/HIV in Manaus, Western Brazilian Amazon, from December 2016 to April 2018. This is a reference public institution for infectious diseases in the Amazonas State, with ~150 beds available for hospitalization and 7 intensive care unit (ICU) beds, where all cases of TB/HIV coinfection are referred to.

## Study participants

Either gender 18 years or older patients with pulmonary TB diagnosed through positive Xpert-MTB/RIF and hospitalized (at clinician's discretion) for more than 24 hours, were eligible to be included in the study. Patients without HIV, with extrapulmonary TB only, unable to collect respiratory sample, pregnant and lactating women, exposed to quinolones in the last 7 days, and in use of anti-TB drugs for more than 72 hours or in use of anti-TB drugs as second line drugs were not included in the study. Enrolled patients were subsequently excluded if their baseline culture failed to grow Mtb or grew a strain of Mtb that was resistant to any anti-TB drug.

For the sample size calculation, a percentage of 37.5% of hepatotoxicity among the RIPE group and no episodes for RIPENAC was considered [12]. A 1:1 ratio with a power of 80% and a significance level of 95% was used. A total sample size of 36 was estimated.

## Randomization and study treatments

Patients were randomized into **Control Group** or **NAC Group** in a 1:1 ratio using a computer-generated randomization table. The groups received standard anti-TB treatment with RIPE (150 mg, 75 mg, 400 mg, 275 mg), fixed dose tablets combined according to weight, for eight weeks. RIPE was supplied by Farmanguinhos[R], Rio de Janeiro, Brazil. In addition, NAC group received two effervescent tablets containing N-acetylcysteine (Fluimucil[R]) 600 mg bid, for eight weeks, following the same dose used in a preliminary study on the effect of NAC on TB [14]. Tablets were dissolved in water before oral ingestion or administration though the nasoenteral tube.

Patients and involved infectious disease physicians were aware of the treatments, except laboratory team, to whom the study was blinded. While hospitalized, all the medication was administered in a supervised way by the nursing team. After discharge, patients were asked to take anti-TB drugs accordingly, and NAC only until eight weeks was completed. During every visit to the clinics, patients were requested to bring medication packages for tablet counting, as a proxy of adherence. Adherence was considered low when patients did not take the medication for more than seven consecutive days. The study was registered at Clinicaltrials.gov. (https://clinicaltrials.gov/ct2/show/NCT03281226)

## Study procedures

All participants underwent a baseline clinical evaluation, which included physical examination, sputum (spontaneous or induced whenever sputum production was considered insufficient) or tracheal aspirate in unconscious patients, CD4[+] lymphocyte count, viral load, aspartate aminotransferase (AST), alanine aminotransferase (ALT), bilirubins, screening of

concomitant drug exposures and chest radiograph. Safety assessments were performed at baseline and weeks 1, 2, 4, 6 and 8. Additional exams were solicited whenever needed.

Respiratory samples were submitted to smear Ziehl–Nielsen staining technique, Xpert-MTB/RIF for Mtb, and sown in liquid culture BACTEC MGIT™ 960 and solid culture Löwenstein-Jensen. Xpert-MTB/RIF for Mtb, even being more expensive, was used as inclusion criterion because of its higher sensitivity [16].

## Study outcomes

The primary endpoint was clinical and laboratorial safety, and tolerability. Radiology alterations recovery, respiratory specimen culture conversion rate on liquid and solid media at the end of eight weeks of treatment were secondary endpoints.

## Definitions

**Culture conversion and rate of culture conversion.** We defined culture conversion as the first negative respiratory sample (sputum or tracheal aspirate) cultures on liquid or solid media, without an intervening positive culture. Negative cultures followed by contaminated cultures were also regarded as culture conversion. Culture conversion was also defined as a case where the participant could not expectorate after one negative sputum culture. The rate of culture conversion was defined as the time elapsed from day 1 to the first negative culture [17].

**Radiology assessment.** Chest radiography was performed at baseline and week 8. Comparative assessment was performed by a single specialist in radiology, blinded to the group of enrollment, which evaluated both exams and classified them as: (1) improvement or no change, or (2) worsening.

**Hepatotoxicity.** Hepatotoxicity was defined as ALT and/or AST increased more than 3 times the upper limit of normal range with the presence of hepatitis symptoms, or increased up to 5 times the upper limit of normal range in the absence of symptoms or total levels of bilirubinemia greater than twice the upper normal limit, as described elsewhere [12]. Reference values adopted were 38 UI/mL (AST), 44 UI/mL (ALT) and 1.3 mg/dL (bilirubins). All patients were tested for HBsAg and anti-HCV.

**Adverse events.** Adverse events were graded according to the modified toxicity events criteria of the *National Institute of Allergy and Infectious Diseases*, Division of AIDS (DAIDS) Table for Grading the Severity of Adult and Pediatric Adverse Event (Corrected Version 2.1, July 2017) [18].

## Statistical analysis

All analyses were performed according to the intention-to-treat principle. Differences in categorical variables were tested using Fisher's exact test. Univariate log-binomial generalized linear regression with respective 95% confidence intervals (CI) was used to estimate relative risks (RR) in order to assess associations with the major outcomes of the study. P-values < .05 were considered statistically significant. The statistical analyses were performed using Stata 13.0.

## Results

Between December 2016 and April 2018, 162 participants were assessed for eligibility, and 50 underwent randomization (Fig 1). Out of those, 21 were included in the Control Group and 18 in the NAC Group. Demographic and clinical characteristics of participants were similar between the study arms, except that more males were included in the control arm (Table 1). Overall, most of the included patients had CD4+ lymphocyte counts under 200 cells/mm$^3$.

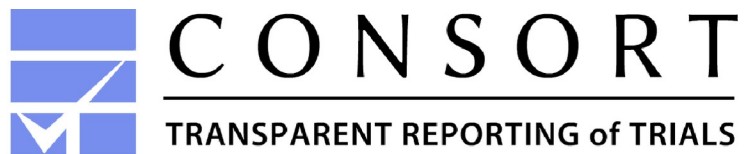

## CONSORT 2010 Flow Diagram

**Enrollment**

127 patients positive for *Mtb* (Xpert-MTB/RIF) in the service were assessed for eligibity

77 were not included:

23 were HIV negative
35 were not hospitalized
8 had rifampicin-resistant *Mtb*
1 used quinolone in the last 7 days
8 unable to be followed-up
2 were unable to produce sputum

Randomized (n=50)

**Allocation**

Allocated to **Control Arm** (n=25)

Allocated to **Intervention Arm** (n=25)

**Follow-Up**

Lost to follow-up (n=4)

2 did not have positive culture at baseline
2 had drug resistant tuberculosis detected

Lost to follow-up (n=7)

3 did not have positive culture at baseline
3 had drug resistant tuberculosis detected
1 withdrawal of consent

**Analysis**

21 were included in the intention-to-treat analysis

18 were included in the intention-to-treat analysis

**Fig 1. Flowchart of eligible, randomized and enrolled patients in the study.**

**Table 1. Baseline characteristics of patients enrolled in both groups.**

| Characteristic | Control Group (n = 21) | NAC Group (n = 18) | P-value |
|---|---|---|---|
| | n (%) | n (%) | |
| **Male sex** | 19 (90.5) | 10 (55.6) | **.025** |
| **Age group, in years** | | | .972 |
| <25 | 4 (19.1) | 4 (22.2) | |
| 25–34 | 6 (28.6) | 4 (22.2) | |
| 35–45 | 8 (38.1) | 7 (38.9) | |
| >45 | 3 (14.3) | 3 (16.7) | |
| **CD4$^+$ lymphocyte count** | | | .336 |
| <50 cells/mm$^3$ | 6 (28.6) | 9 (50) | |
| 50–199 cells/mm$^3$ | 12 (57.1) | 8 (44.4) | |
| ≥200 cells/mm$^3$ | 3 (14.3) | 1 (5.6) | |
| **Viral load** | | | .873 |
| ≤ 400 copies/mL | 5 (23.8) | 4 (22.2) | |
| 401–3,000 copies/mL | 1 (4.8) | 1 (5.6) | |
| 3,001–10,000 copies/mL | 1 (4.8) | 0 (0.0) | |
| 10,001–100,000 copies/mL | 3 (14.3) | 4 (22.2) | |
| >100,000 copies/mL | 11 (52.4) | 9 (50) | |
| **Hemoglobin<8g/dL** | 4 (19.1) | 5 (27.8) | .706 |
| **Albumin≤2.7mg/dL** | 4 (19.1) | 8 (44.5) | .163 |
| **Time between TB diagnosis and ART in naïve patients** | | | .549 |
| < 2 weeks | 5 (55.6) | 4 (50) | |
| 2 weeks—8 weeks | 3 (33.3) | 4 (50) | |
| > 8 weeks | 1 (11.1) | 0 (0.0) | |
| **Concurrent opportunistic infection** | | | .196 |
| No | 16 (76.2) | 10 (55.6) | |
| Yes | 5 (23.8) | 8 (44.4) | |
| **Concurrent extrapulmonary TB** | 14 (66.7) | 11 (61.1) | .750 |
| **Disseminated TB (≥2 sites)** | 3 (14.3) | 4 (22.2) | .683 |
| **Years since HIV diagnosis** | | | .471 |
| <1 year | 15 (83.4) | 13 (62) | |
| >1 year | 8 (38.1) | 3 (16.7) | |
| **ICU hospitalization at enrollment** | 4 (19.1) | 0 (0.0) | . . . |

ART: antiretroviral therapy

Overall, TB resistance was seen in 5 out of 50 enrolled patients (10%). Geometric means of viral load in the control group was 4.133 copies/mL and in the NAC group 4.569 copies/mL.

As per protocol, no patient in the NAC Group had low adherence to NAC. Table 2 shows similar types of adverse events seen in both groups, and Table 3 shows the similar grading. No patient was positive for HBV or HCV.

Fig 2 shows the major outcomes related to the efficacy of the NAC arm. No differences were seen. Fig 3 details the quantification of ALT over the weeks, reinforcing that no change was seen between the groups.

## Discussion

This trial conducted with TB/HIV coinfected hospitalized patients aimed to estimate whether the use of NAC together with RIPE was not unsafe.

**Table 2. Major adverse events seen in both groups.**

| Adverse event | Control Group (n = 21) | NAC Group (n = 18) | |
|---|---|---|---|
| **Gastrointestinal disorders** | | | |
| Gastric fullness | 0 | 1 (5.5) | . . . |
| Dysphagia | 0 | 2 (11.1) | . . . |
| Nausea | 1 (4.7) | 3 (16.6) | 0.345 |
| Vomiting | 2 (9.5) | 4 (22.2) | 0.414 |
| Hepatotoxicity | 7 (33.3) | 10 (55.5) | 0.562 |
| **Respiratory disorders** | | | |
| Dyspnea | 0 | 1 (5.5) | . . . |
| **Other disorders** | | | |
| Pyrosis | 0 | 1 (5.5) | . . . |
| Pruritus | 0 | 1 (5.55) | . . . |
| Rash | 1 (4.7) | 0 | . . . |

NS: Non-Significant

Hospitalized patients, used here as a proxy of clinical severity, were the targeted population because of their increased likelihood of evolving to death, and therefore, more prone to adhere to adjunctive therapy. Non-severe HIV/TB patients are already in use of many drugs simultaneously, and any adjunctive therapy would compromise adherence if a major benefit is not clearly seen by the patient. Therefore, we believe that adjunctive therapy in HIV/TB coinfection must be designed to give priority to more complicated patients. That requests that safety and efficacy studies are performed in this population since the very beginning of the evidence generation process.

For non-severe TB patients, NAC has been pursued as an adjunctive drug to decrease hepatotoxicity, a problem that still persists in ~25% of patients, impacting adherence to RIPE and TB cure, ultimately [19]. The only clinical trial in which NAC was concurrently used in pulmonary TB, NAC was significantly associated to faster sputum negativity, improved radiological response, weight, serum glutathione peroxidase level, and amelioration of the deregulated immune response [20].

In PLWH, up to 30% of the patients experience hepatotoxicity, HIV infection apparently being one predisposing factor [21]. However, not only hepatotoxicity is an expected effect of NAC adjunctive therapy in this population, but also culture conversion, used routinely as a

**Table 3. Grading of adverse events seen in both groups.**

| Adverse event | | Control Group (n = 21) | | NAC Group (n = 18) | | |
|---|---|---|---|---|---|---|
| | | Number of events | Number of participants (%) | Number of events | Number of participants (%) | p-value |
| No events | | . . . | 6 (28.6) | . . . | 2 (11.1) | 0.427 |
| Any event, except death | | 33 | 13 (61.9) | 41 | 14 (77.8) | 0.322 |
| Grade 1 | | 15 | 10 (47.6) | 18 | 12 (55.6) | 0.201 |
| Grade 2 | | 10 | 7 (33.3) | 12 | 7 (38.9) | 0.750 |
| Grade 3 | | 4 | 3 (14.3) | 10 | 5 (27.8) | 0.682 |
| Grade 4 | | 4 | 2 (9.5) | 1 | 1 (5.6) | 1 |
| Death | | . . . | 4 (19.1) | . . . | 5 (27.8) | 0.706 |

NS: Non-Significant

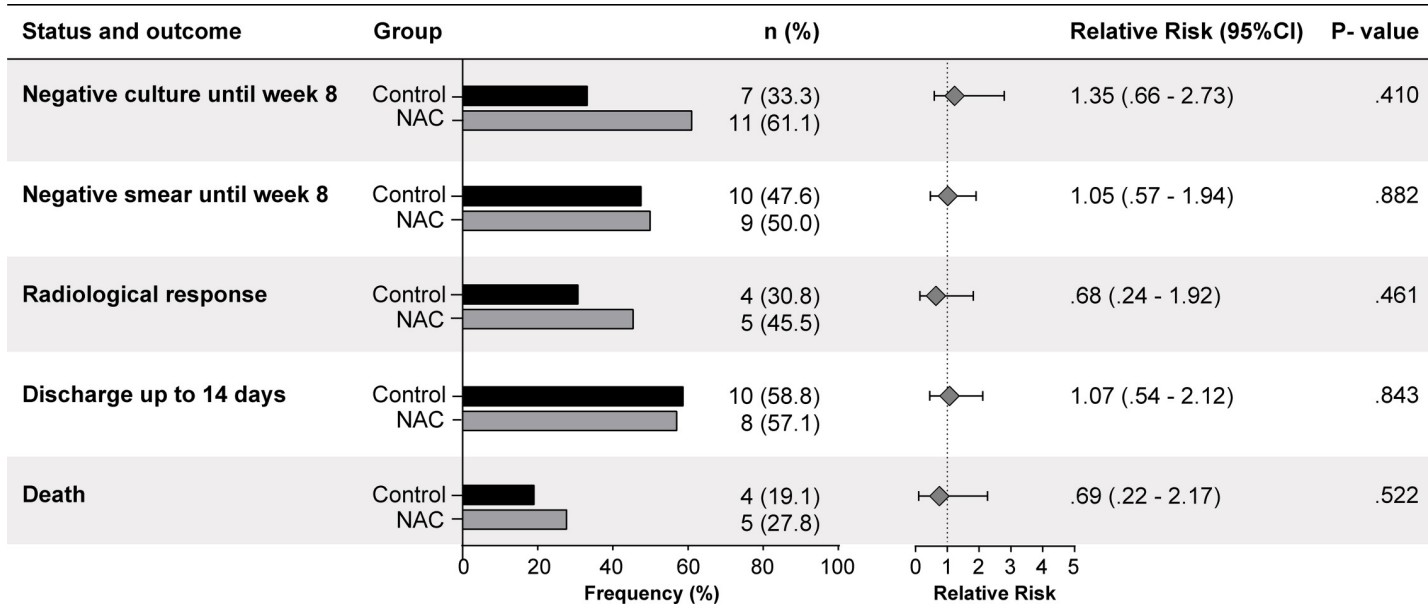

**Fig 2. Major outcomes and respective 95% confidence intervals.** P-values refer to RR estimates.

marker of TB clinical recovery. This is critical in patients with immune deficiency, and increased risk of death. To date, however, no proof of concept study has been performed targeting this high-risk population of PLWH and TB coinfection. The selected dose of NAC was chosen based on the lack of immunological effect in PLWH using 600 mg qd [22], and the promising dose of 600 mg bid in TB patients [20].

Safety was the major outcome of interest in our study, considering that TB/HIV hospitalized patients are a special group of subjects under enormous stress, with very low CD4+ lymphocyte counts, and in use of many drugs with potential interactions. During the eight weeks in which NAC was used in one of the arms, adverse events were seen in similar proportions in both arms, gastrointestinal events being the most frequent (Table 2). When the total number of events and grading were assessed, no significant findings were seen, evidencing that NAC adjunctive to RIPE is not unsafe as compared to RIPE by itself in coinfected patients. No trends in decrease of hepatotoxicity in patients using NAC was seen, suggesting that oxidative stress only partially explains liver damage in these patients in use of RIPE.

TB resistance, even not being a major focus of this work, was found in 10% of the enrolled patients, which is pretty much the same percentage as seen in other similar settings in Brazil [23].

Major limitation of the trial was the reduced sample size, which did not allow for a more robust statistical analysis. However, the universe of eligible patients seen in such a reference unit is not much bigger, and the whole study recruited patients over 16 months. A multicentric approach will certainly be needed in further phase III studies. It is also known that more males present HIV/TB coinfection in Brazil [24], but in this randomized trial, Table 1 shows that more males were enrolled in the Control Group, what may be explained by chance and the small sample size. No clear bias was considered. Noteworthy to say that results found here may not be extrapolated to non-severe patients seen in the outpatient clinics. Likewise, results are not applicable to multidrug-resistant Mtb [25], a condition in which adjunctive therapies are also needed. Some patients were already hospitalized in the ICU during enrollment, and therefore, ICU hospitalization as an endpoint had limitation in the analyses. Data from our group

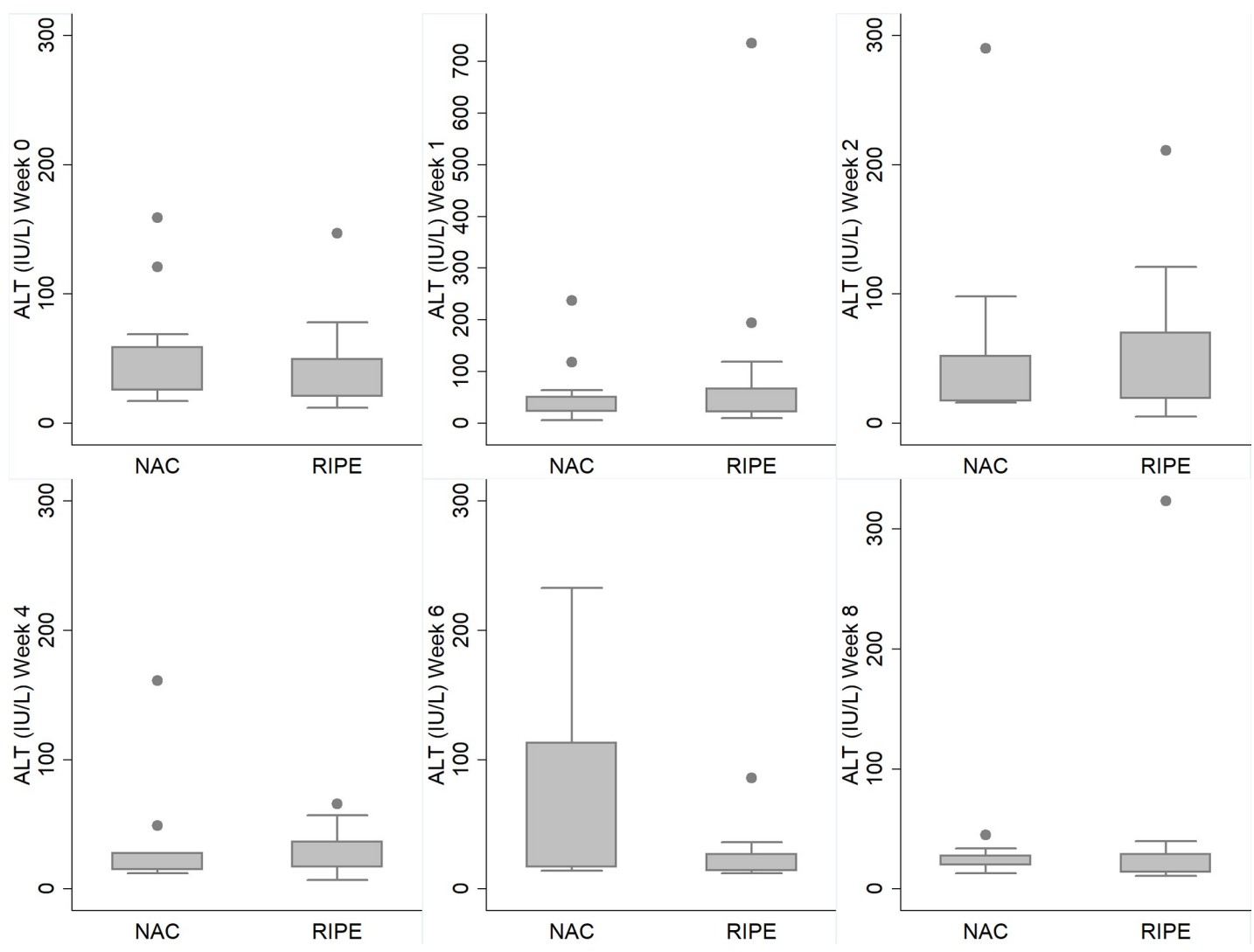

**Fig 3. ALT levels between control and NAC groups over the weeks of the follow-up.**

show high early mortality rate amongst TB/HIV coinfected ICU patients. The factors predictive of mortality in this population were invasive mechanical ventilation, hypoalbuminemia, and severe immunosuppression [26]. Indigenous population was also excluded from the analysis, but represents a major burden of the disease in the Brazilian Amazon [27,28].

As the use of NAC in the HIV/TB population seems promising in terms of safety, our results indicate that RIPE plus NAC regimen is suitable for a larger phase III trial. It is worth mentioning that NAC has a well-known safety profile safety, even in much higher doses [29], is tolerable in pregnant women, is quite economically affordable, and requests no major medical supervision during the administration of oral presentations, mostly flavored. During oral administration, deacetylation reaction of NAC happens while passing along the small intestine as well as liver, thus its bioavailability is only 4–10% decreased [30]. The ongoing TB-SEQUEL cohort study (ClinicalTrials.gov Identifier: NCT03702738) aims to evaluate similar endpoints, using a higher dose of 1,200 mg bid of NAC in patients with TB, with and without HIV coinfection. Therefore, in the near future, more evidence will be generated to support the use of

this safe drug in coinfected patients, still a major contributor to mortality in developing countries.

## Supporting information

**S1 Checklist. CONSORT 2010 checklist of information to include when reporting a randomised trial**∗**.**
(DOC)

**S1 Dataset.**
(XLSX)

**S1 File.**
(PDF)

## Acknowledgments

The authors would like to thank all patients and the *Fundação de Medicina Tropical Dr Heitor Vieira Dourado* Tuberculosis Laboratory staff. This manuscript is dedicated to *Maísa Safe Lacerda*, a 'secondary endpoint' of this trial.

## Author Contributions

**Conceptualization:** Izabella Picinin Safe, Marcus Vinícius Guimarães Lacerda, Alexandra Brito Souza, Francisco Beraldi-Magalhães, Wuelton Marcelo Monteiro, Vanderson Souza Sampaio, Eduardo P. Amaral, Bruno B. Andrade, Marcelo Cordeiro-Santos.

**Data curation:** Marcus Vinícius Guimarães Lacerda, Vitoria Silva Printes, Adriana Ferreira Praia Marins, Amanda Lia Rebelo Rabelo, Amanda Araújo Costa, Michel Araújo Tavares, Jaquelane Silva Jesus, Alexandra Brito Souza, Francisco Beraldi-Magalhães, Cynthia Pessoa Neves, Wuelton Marcelo Monteiro, Vanderson Souza Sampaio, Bruno B. Andrade, Marcelo Cordeiro-Santos.

**Formal analysis:** Marcus Vinícius Guimarães Lacerda, Wuelton Marcelo Monteiro, Vanderson Souza Sampaio, Bruno B. Andrade, Marcelo Cordeiro-Santos.

**Funding acquisition:** Marcus Vinícius Guimarães Lacerda, Marcelo Cordeiro-Santos.

**Investigation:** Marcus Vinícius Guimarães Lacerda, Bruno B. Andrade, Marcelo Cordeiro-Santos.

**Methodology:** Marcus Vinícius Guimarães Lacerda, Francisco Beraldi-Magalhães, Wuelton Marcelo Monteiro, Bruno B. Andrade.

**Project administration:** Marcus Vinícius Guimarães Lacerda.

**Resources:** Marcus Vinícius Guimarães Lacerda.

**Supervision:** Marcus Vinícius Guimarães Lacerda, Marcelo Cordeiro-Santos.

**Visualization:** Marcus Vinícius Guimarães Lacerda, Bruno B. Andrade.

**Writing – original draft:** Izabella Picinin Safe, Marcus Vinícius Guimarães Lacerda.

**Writing – review & editing:** Izabella Picinin Safe, Marcus Vinícius Guimarães Lacerda, Wuelton Marcelo Monteiro, Renata Spener Gomes, Bruno B. Andrade, Marcelo Cordeiro-Santos.

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
