## [Decision Letter · Decision Letter 0]

2 Jan 2020

PONE-D-19-21939

Safety and efficacy of N-acetylcysteine in hospitalized patients with HIV-associated tuberculosis: an open-label, randomized, phase II trial (RIPENACTB Study)

PLOS ONE

Dear Dr. Lacerda,

Thank you for submitting your manuscript to PLOS ONE. After careful consideration, we feel that it has merit but does not fully meet PLOS ONE’s publication criteria as it currently stands. Therefore, we invite you to submit a revised version of the manuscript that addresses the points raised during the review process.

We would appreciate receiving your revised manuscript by Feb 16 2020 11:59PM. To enhance the reproducibility of your results, we recommend that if applicable you deposit your laboratory protocols in protocols.io, where a protocol can be assigned its own identifier (DOI) such that it can be cited independently in the future. For instructions see: http://journals.plos.org/plosone/s/submission-guidelines#loc-laboratory-protocols

We look forward to receiving your revised manuscript.

Kind regards,

Delia Goletti, M.D., Ph.D.

Academic Editor

PLOS ONE

Journal Requirements:

https://www.thelancet.com/journals/lanhiv/article/PIIS2352-3018(19)30154-7/fulltext

https://academic.oup.com/cid/advance-article-abstract/doi/10.1093/cid/ciz152/5365529?redirectedFrom=fulltext

The text that needs to be addressed is in the Introduction section.

In your revision ensure you cite all your sources (including your own works), and quote or rephrase any duplicated text outside the methods section. Further consideration is dependent on these concerns being addressed.

3. Thank you for submitting your clinical trial to PLOS ONE and for providing the name of the registry and the registration number. The information in the registry entry suggests that your trial was registered after patient recruitment began. PLOS ONE strongly encourages authors to register all trials before recruiting the first participant in a study.

1) your reasons for your delay in registering this study (after enrolment of participants started);

2) confirmation that all related trials are registered by stating: “The authors confirm that all ongoing and related trials for this drug/intervention are registered”.

Please also ensure you report the date at which the ethics committee approved the study as well as the complete date range for patient recruitment and follow-up in the Methods section of your manuscript.

Additional Editor Comments (if provided):

the manuscript is interesting. It opens new insights in the treatment of HIV-TB patients.

Please, answer the referees' major points regarding:

1. safety issues

2. lenght of recruitment of the patients

Reviewers' comments:

Reviewer's Responses to Questions

**Comments to the Author**

1. Is the manuscript technically sound, and do the data support the conclusions?

Reviewer #1: No

Reviewer #2: Partly

Reviewer #3: Yes

2. Has the statistical analysis been performed appropriately and rigorously? 

Reviewer #1: No

Reviewer #2: No

Reviewer #3: Yes

3. Have the authors made all data underlying the findings in their manuscript fully available?

Reviewer #1: Yes

Reviewer #2: No

Reviewer #3: Yes

4. Is the manuscript presented in an intelligible fashion and written in standard English?

Reviewer #1: Yes

Reviewer #2: No

Reviewer #3: Yes

5. Review Comments to the Author

Reviewer #1: 1. My main criticism of this manuscript is that you have not shown that the treatment condition is safe, only that it is not unsafe. You are arguing that not rejecting the null hypothesis implies the groups are similar, i.e., absence of evidence is evidence of absence. By this logic, I could test two people in each group and conclude the treatment group is "safe". This is a huge oversight and, as a statistician, very disappointing to see this. To conclude these groups are "similar", you would need to perform equivalence testing or noninferiority testing (provided you are able to set a margin for noninferiority).

2. Given the low sample sizes of the groups, I would recommend not using asymptotic tests, such as Pearson's Chi-squared. I would switch to either permutation tests or Fisher's Exact Test. Also, if you use one of those, you can tests for differences between groups with small numbers of events…provided that hypothesis fits with your goals.

3. The same is true for the logistic regression as I would recommend exact logistic regression. I also don't entirely understand the purpose of the regression models as all that I see mentioned in the text is the p-value. There seems to be no interest in the odds ratios, except in the figures.

4. Also, while the small sample size is a limitation, that is made all the much worse by using binary outcomes. Certainly, some are binary by nature, but some like hepatotoxicity could have the quantitative values analyzed. Research suggests the loss of power after dichotomizing is large (e.g., https://doi.org/10.1002/pst.331). There may be industry standards about how to cut an outcome. Fair, but this seems like a really good opportunity to analyze the quantitative values.

Reviewer #2: 1. Reference, for instance, reference 1, can be more updated.

2. Your underlying data regarding your manuscript cannot be found from the given URL.

3. Is it necessary to conduct the study in hospitalized patients? As we can have more participants without this criterion. It also quite a surprise to see not many participants in the study given 1.5 years of study duration even Brazil is one of the countries with the highest burden of TB and TB with HIV.

4. The study design was a phase II clinical trial to see the safety of the intervention. One should rather select less severe participants. However, in this study, one of the inclusion criteria was being hospitalized patient which was relatively severe.

5. The section "Results" was far too short. It needs more detail e.g., detail of liver function profiles at weeks 1, 2, 4, 6 and 8.

6. The culture conversion rate can also be calculated using hazard ratio as you mentioned in the definition that "The rate of culture conversion was defined as the time elapsed from day 1 to the first negative culture".

7. In this RCT, you can interpret your findings using relative risk or hazard ratio

8. For Table 1, mean with SD or median with IQR should be provided together with each continuous variable rather than just the stratified values. For viral load, you might consider presenting your information with the logarithm of viral load.

9. You stated to use logistic regression analysis, however, its results did not appear in the results.

10. Your conclusion was over claim for the benefit of NAC given the sample size of less than 40.

11. Given your Figure 1, it is quite interesting to see such a high rate of drug resistance tuberculosis; 5 out of 50 in both arms.

Reviewer #3: Some sentences needs revision to be fully understandable. you do not need to use long and confusing sentences in discussion. Break them to small and clear sentences.

This is a useful study in this field, similar to our study, that is not published yet.

6. PLOS authors have the option to publish the peer review history of their article (what does this mean?). If published, this will include your full peer review and any attached files.

Reviewer #1: No

Reviewer #2: No

Reviewer #3: Yes: Nasser Vahdati-Mashhadian

---

## [Author Response · Author response to Decision Letter 0]

10 Mar 2020

Rebuttal Letter

Dear editor,

We very much appreciate the comments made by the reviewers. Despite the low number of patients in this study, we still believe that it paves the way for further studies on the subject.

Comments are made in the following lines, accordingly.

Additional Editor Comments:

The manuscript is interesting. It opens new insights in the treatment of HIV-TB patients.

Please, answer the referees' major points regarding:

1. safety issues

2. lenght of recruitment of the patients

These issues are addressed through the responses to the reviewers (see next).

Reviewers' comments:

Reviewer's Responses to Questions

Comments to the Author

Reviewer #1: 

1. My main criticism of this manuscript is that you have not shown that the treatment condition is safe, only that it is not unsafe. You are arguing that not rejecting the null hypothesis implies the groups are similar, i.e., absence of evidence is evidence of absence. By this logic, I could test two people in each group and conclude the treatment group is "safe". This is a huge oversight and, as a statistician, very disappointing to see this. To conclude these groups are "similar", you would need to perform equivalence testing or noninferiority testing (provided you are able to set a margin for noninferiority).

Considering that we did not plan the study to be a non-inferiority trial, these margins were not used in principle. We fully agree that ‘safety’ has to be used with care, therefore, we have changed the whole discussion section to address the concern of the reviewer. We agree that NAC group ‘seemed’ only to be not unsafe in our limited sample. Therefore, the limitations of a more robust sample size are compensated somehow by a more cautious discussion and interpretation of our data.

2. Given the low sample sizes of the groups, I would recommend not using asymptotic tests, such as Pearson's Chi-squared. I would switch to either permutation tests or Fisher's Exact Test. Also, if you use one of those, you can test for differences between groups with small numbers of events…provided that hypothesis fits with your goals.

We fully agree with the suggestion, and Fisher’s exact test and new p-values were used, accordingly.

3. The same is true for the logistic regression as I would recommend exact logistic regression. I also don't entirely understand the purpose of the regression models as all that I see mentioned in the text is the p-value. There seems to be no interest in the odds ratios, except in the figures.

We emphasize that no regression models were used in the analyses. We have updated the RR analysis performed through generalized linear regression with log binomial.

4. Also, while the small sample size is a limitation, that is made all the much worse by using binary outcomes. Certainly, some are binary by nature, but some like hepatotoxicity could have the quantitative values analyzed. Research suggests the loss of power after dichotomizing is large (e.g., https://doi.org/10.1002/pst.331). There may be industry standards about how to cut an outcome. Fair, but this seems like a really good opportunity to analyze the quantitative values.

Not only industry, but clinical relevance, according to DAIDS (Division of Aids - NIH). Also, it allows comparisons with most of the published similar trials. Sometimes significant increase in quantitative values have absolutely no clinical relevance but may flag unreasonable concerns. However, we fully agree that hepatotoxicity is a key safety concern and therefore, we have analyzed it quantitatively and presented in the Results section.

Reviewer #2:

1. Reference, for instance, reference 1, can be more updated.

Indeed, due to the delay in the responses by PLoS One, a new 2019 version of the TB global report is available and it was updated accordingly, as well as the respective cited data.

2. Your underlying data regarding your manuscript cannot be found from the given URL.

URL is facing recent issues. Therefore, we have decided to submit the databank as supplementary material.

3. Is it necessary to conduct the study in hospitalized patients? As we can have more participants without this criterion. It also quite a surprise to see not many participants in the study given 1.5 years of study duration even Brazil is one of the countries with the highest burden of TB and TB with HIV.

Actually, as explained in the MS, our major goal was to test the safety of NAC amongst patients who would benefit most from this additional adjunctive drug, i.e., complicated hospitalized patients with TB-HIV. That is the target population we believe will benefit most from adding a new drug in the very busy drug regimen these patients already face routinely. 

Ideally this could have been a multicentric study, however, more expensive. Brazil and the Brazilian Amazon actually present large TB cases numbers, but not necessarily in HIV patients. In Brazil, only 0.5% of general population is positive for HIV. Hospitalized patients are only a small sub-sample of those.

4. The study design was a phase II clinical trial to see the safety of the intervention. One should rather select less severe participants. However, in this study, one of the inclusion criteria was being hospitalized patient which was relatively severe.

As stated before, we have addressed this phase II clinical trial to the targeted population we expect to benefit most from the novelty in the future. There might be different views regarding this target product profile (TPP), but with all the respect, we do not envision all patients with TB-HIV using NAC for 8 weeks, considering that compliance is already very low in many endemic areas.

5. The section "Results" was far too short. It needs more detail e.g., detail of liver function profiles at weeks 1, 2, 4, 6 and 8.

A new graph addressing ALT levels per week amongst groups was added.

6. The culture conversion rate can also be calculated using hazard ratio as you mentioned in the definition that "The rate of culture conversion was defined as the time elapsed from day 1 to the first negative culture".

We agree with the reviewer, and the figure was updated accordingly. Data do not refer to week 8 negative culture, but negative culture until week 8, which is different.

7. In this RCT, you can interpret your findings using relative risk or hazard ratio.

The reviewer is correct, relative risk (RR) is a better analysis in a prospective study, such as a clinical trial. Figure 1 was updated accordingly.

8. For Table 1, mean with SD or median with IQR should be provided together with each continuous variable rather than just the stratified values. For viral load, you might consider presenting your information with the logarithm of viral load.

That would highly increase the size of the MS, not necessarily with useful information. However we have presented viral load data as geometric means, instead of arithmetic means, considering the point raised by the reviewer.

9. You stated to use logistic regression analysis, however, its results did not appear in the results.

We have changed now data for relative risk, therefore, regression analysis was not used anymore.

10. Your conclusion was over claim for the benefit of NAC given the sample size of less than 40.

We were more cautious in that statement and changes were made accordingly.

11. Given your Figure 1, it is quite interesting to see such a high rate of drug resistance tuberculosis; 5 out of 50 in both arms.

In hospitalized TB/HIV patients in Brazil, recent similar data point to 14% of resistance in this population (Microbiol Insights. 2018 Nov 27;11: Human Immunodeficiency Virus and Tuberculosis Coinfection in a Tertiary Hospital in Southern Brazil: Clinical Profile and Outcomes. Teixeira et al). The reference was added to the Discussion section and the valuable point made.

Reviewer #3: 

Some sentences needs revision to be fully understandable. you do not need to use long and confusing sentences in discussion. Break them to small and clear sentences.

Changes were made accordingly.

This is a useful study in this field, similar to our study, that is not published yet.

That is great to hear. Thanks for sharing.

---

## [Decision Letter · Decision Letter 1]

14 May 2020

PONE-D-19-21939R1

Safety and efficacy of N-acetylcysteine in hospitalized patients with HIV-associated tuberculosis: an open-label, randomized, phase II trial (RIPENACTB Study)

PLOS ONE

Dear dr LACERDA, deep apologies for the delay.

Thank you for submitting your manuscript to PLOS ONE. After careful consideration, we feel that it has merit but does not fully meet PLOS ONE’s publication criteria as it currently stands. Therefore, we invite you to submit a revised version of the manuscript that addresses the points raised during the review process.

ACADEMIC EDITOR: 

the manuscript has been revised by additional two statisticians.

Few comments are needed:

 1- Clarify that the adverse event RATES were similar in the two arms.

2- Statistical Analysis Section:

A. Differences in categorical variables were TESTED using Fisher’s exact test.

B. This statement is unclear, “generalized linear regression with log binomial.” More fully explain the statistical methods associated with the general linear regression model.

C. Remove “Differences with” in the sentence “Differences with p-values < 0.05 were  considered statistically significant.

3- Replace NS with numeric p-values.

4- Figure 2: Indicate the type of test used to generate the p-value.

5- State and justify the study’s target sample size with a pre-study statistical power calculation. thank you for being patient. 

We would appreciate receiving your revised manuscript by Jun 28 2020 11:59PM. To enhance the reproducibility of your results, we recommend that if applicable you deposit your laboratory protocols in protocols.io, where a protocol can be assigned its own identifier (DOI) such that it can be cited independently in the future. For instructions see: http://journals.plos.org/plosone/s/submission-guidelines#loc-laboratory-protocols

We look forward to receiving your revised manuscript.

Kind regards,

Delia Goletti, M.D., Ph.D.

Academic Editor

PLOS ONE

Reviewers' comments:

Reviewer's Responses to Questions

**Comments to the Author**

1. If the authors have adequately addressed your comments raised in a previous round of review and you feel that this manuscript is now acceptable for publication, you may indicate that here to bypass the “Comments to the Author” section, enter your conflict of interest statement in the “Confidential to Editor” section, and submit your "Accept" recommendation.

Reviewer #3: (No Response)

Reviewer #4: (No Response)

2. Is the manuscript technically sound, and do the data support the conclusions?

Reviewer #3: Yes

Reviewer #4: Partly

3. Has the statistical analysis been performed appropriately and rigorously? 

Reviewer #3: Yes

Reviewer #4: Yes

4. Have the authors made all data underlying the findings in their manuscript fully available?

Reviewer #3: Yes

Reviewer #4: Yes

5. Is the manuscript presented in an intelligible fashion and written in standard English?

Reviewer #3: Yes

Reviewer #4: Yes

6. Review Comments to the Author

Reviewer #3: (No Response)

Reviewer #4: A phase II randomized two-arm controlled clinical trial was conducted to assess the safety and tolerability of N-acetylcysteine (NAC) in HIV patients with tuberculosis. The adverse event rates were similar in the two arms, and no evidence was found to refute that NAC was unsafe.

Minor revisions:

1- Clarify that the adverse event RATES were similar in the two arms.

2- Statistical Analysis Section:

A. Differences in categorical variables were TESTED using Fisher’s exact test.

B. This statement is unclear, “generalized linear regression with log binomial.” More fully explain the statistical methods associated with the general linear regression model.

C. Remove “Differences with” in the sentence “Differences with p-values < 0.05 were considered statistically significant.

3- Replace NS with numeric p-values.

4- Figure 2: Indicate the type of test used to generate the p-value.

5- State and justify the study’s target sample size with a pre-study statistical power calculation.

7. PLOS authors have the option to publish the peer review history of their article (what does this mean?). If published, this will include your full peer review and any attached files.

Reviewer #3: No

Reviewer #4: No

---

## [Author Response · Author response to Decision Letter 1]

10 Jun 2020

Dear Editor,

We appreciate the opportunity to clarify the points raised by the reviewer and believe that the manuscript has improved significantly.

Reviewer #4: A phase II randomized two-arm controlled clinical trial was conducted to assess the safety and tolerability of N-acetylcysteine (NAC) in HIV patients with tuberculosis. The adverse event rates were similar in the two arms, and no evidence was found to refute that NAC was unsafe.

Minor revisions:

1- Clarify that the adverse event RATES were similar in the two arms.

Response: The sentence was adjusted accordingly in the abstract.

2- Statistical Analysis Section:

A. Differences in categorical variables were TESTED using Fisher’s exact test.

Response: The sentence was adjusted accordingly.

B. This statement is unclear, “generalized linear regression with log binomial.” More fully explain the statistical methods associated with the general linear regression model.

Response: The sentence was rephrased to clarify this point.

“Univariate log-binomial generalized linear regression with respective 95% confidence intervals (CI) was used to estimate relative risks (RR) in order to assess associations with the major outcomes of the study”.

C. Remove “Differences with” in the sentence “Differences with p-values < 0.05 were considered statistically significant.

Response: The sentence was adjusted accordingly.

3- Replace NS with numeric p-values.

Response: Replaced accordingly.

4- Figure 2: Indicate the type of test used to generate the p-value.

Response: The following sentence was added to the graph legend: “P-values refer to RR estimates” to clarify.

5- State and justify the study’s target sample size with a pre-study statistical power calculation.

Response: A paragraph including sample size calculation was added to the manuscript Study participants section as follows: “For the sample size calculation, a percentage of 37.5% of hepatotoxicity among the RIPE group and no episodes for RIPENAC was considered [12]. A 1:1 ratio with a power of 80% and a significance level of 95% was used. A total sample size of 36 was estimated.

---

## [Editor Report · Decision Letter 2]

16 Jun 2020

Safety and efficacy of N-acetylcysteine in hospitalized patients with HIV-associated tuberculosis: an open-label, randomized, phase II trial (RIPENACTB Study)

PONE-D-19-21939R2

Dear Dr. Marcus VG Lacerda, 

We’re pleased to inform you that your manuscript has been judged scientifically suitable for publication and will be formally accepted for publication once it meets all outstanding technical requirements.

Kind regards,

Delia Goletti, M.D., Ph.D.

Academic Editor

PLOS ONE

Additional Editor Comments (optional):

The authors answered the questions raised.
---

## [Editor Report · Acceptance letter]

18 Jun 2020

PONE-D-19-21939R2 

Safety and efficacy of N-acetylcysteine in hospitalized patients with HIV-associated tuberculosis: an open-label, randomized, phase II trial (RIPENACTB Study) 

Dear Dr. Lacerda:

I'm pleased to inform you that your manuscript has been deemed suitable for publication in PLOS ONE. Congratulations! Your manuscript is now with our production department. 

Kind regards, 

on behalf of

Dr. Delia Goletti 

Academic Editor

PLOS ONE